# Van Allen Probes observation of plasmaspheric hiss modulated by injected energetic electrons

Run Shi[1], Wen Li[1], Qianli Ma[2,1], Seth G. Claudepierre[3], Craig A. Kletzing[4], William S. Kurth[4], George B. Hospodarsky[4], Harlan E. Spence[5], Geoff D. Reeves[6], Joseph F. Fennell[3], J. Bernard. Blake[3], Scott A. Thaller[7], and John R. Wygant[7]

[1]Center for Space Physics, Boston University, Boston, Massachusetts, USA.

[2]Department of Atmospheric and Oceanic Sciences, University of California, Los Angeles, Los Angeles, California, USA.

[3]Space Science Department, The Aerospace Corporation, El Segundo, California, USA.

[4]Department of Physics and Astronomy, University of Iowa, Iowa City, Iowa, USA.

[5]Institute for the Study of Earth, Oceans, and Space, University of New Hampshire, Durham, New Hampshire, USA.

[6]Space Science and Applications Group, Los Alamos National Laboratory, Los Alamos, New Mexico, USA.

[7]School of Physics and Astronomy, University of Minnesota, Twin Cities, Minneapolis, Minnesota, USA.

Corresponding author:
Run Shi
Center for Space Physics, Boston University, Boston, Massachusetts, USA
runs@bu.edu

**Key points**

1. Clear evidence is provided for local amplification of plasmaspheric hiss by anisotropic
electron distributions
2. Hiss wave intensity variation is well correlated with injected electron flux modulation
3. The modulation of injected electron fluxes is correlated with ULF wave fluctuations

## Abstract

Plasmaspheric hiss was observed by Van Allen Probe B in association with energetic electron injections in the outer plasmasphere. The energy of injected electrons coincides with the minimum resonant energy calculated for the observed hiss wave frequency. Interestingly, the variations of hiss wave intensity, electron flux, and ULF wave intensity exhibit remarkable correlations, while plasma density is not correlated with any of these parameters. Our study provides direct evidence for the first time that the injected anisotropic electron population, which is modulated by ULF waves, modulates the hiss intensity in the outer plasmasphere. This also implies that plasmaspheric hiss observed by Van Allen Probe B in the outer plasmasphere ($L > {\sim}5.5$) is locally amplified. Meanwhile, Van Allen Probe A observed hiss emission at lower $L$ shells ($< 5$), which was not associated with electron injections but primarily modulated by the plasma density. The features observed by Van Allen Probe A suggest that the observed hiss deep inside the plasmasphere may have propagated from higher $L$ shells.

## 1. Introduction

Plasmaspheric hiss plays an important role in the loss of energetic electrons within the plasmasphere and in high-density plumes [*Lyons et al.*, 1972; *Lyons and Thorne*, 1973; *Albert*, 2005; *Meredith et al.*, 2007, 2009; *Summers et al.*, 2008; *Ni et al.*, 2013; *Breneman et al.*, 2015; *Li et al.,* 2015a; *Ma et al.,* 2016]. However, the generation mechanisms of plasmaspheric hiss remain under active research. Three mechanisms have received the

most intense attention to explain the generation of plasmaspheric hiss, including in situ
growth of waves [*Thorne et al.*, 1979; *Church and Thorne*, 1983], lightning generated
whistlers [*Green et al.*, 2005], and whistler mode chorus waves as an "embryonic source"
[*Bortnik et al.* 2008, 2009; *Chen et al.* 2012a, 2012b]. Although wave power above 2–3
kHz from lightning-generated whistlers shows some correlation with hiss waves [*Green et*
*al.*, 2005], the waves below 1 kHz, which contain the majority of hiss wave power, are
independent of lightning flash rate [*Meredith et al.,* 2006]. The in situ growth of waves
inside the plasmasphere was shown to be inadequate to account for the observational level
(~20 dB) [*Huang et al.*, 1983]; in response, *Church and Thorne* [1983] suggested that an
"embryonic source" is required to lead to the observed wave intensity. Recent studies based
on ray tracing simulation [*Bortnik et al.*, 2008] have demonstrated that chorus waves from
the distant magnetosphere can propagate into the plasmasphere and act as an embryonic
source for the hiss wave generation. Furthermore, ray tracing simulations [*Chen et al.*,
2012a] suggested that the majority of hiss formation is caused by chorus emission
originating within ~3 $R_E$ from the plasmapause. This model has successfully explained the
observed frequency spectrum and spatial distribution of the observed hiss over the typical
hiss frequency range from 100 Hz to several kHz. A number of observational studies
[*Bortnik et al.*, 2009; *Wang et al.*, 2011; *Meredith et al.* 2013; *Li et al.*, 2015b] have shown
good correlations between chorus and plasmaspheric hiss and suggested that chorus plays
an important role in hiss wave intensification.
Van Allen Probes recently detected unusually low frequency hiss emissions with wave
power extending well below 100 Hz [*Li et al.,* 2013]. The low frequency hiss was
demonstrated to cause more efficient loss of high energy electrons (from ~50 keV to a few
MeV) due to its stronger pitch angle scattering rates compared to normal hiss [*Ni et al.*,
2014; *Li et al.*, 2015a]. Such low frequency hiss is unlikely to be a result of propagation of
chorus waves from a more distant region because embryonic chorus waves at the same
frequency [*Bortnik et al.*, 2008] would need to originate from unrealistically high $L$ shells
[*Li et al.,* 2015b]. Therefore, these low frequency hiss waves were suggested to be
generated in the outer plasmasphere on the dayside through local amplification [*Li et al.*,
2013; *Chen et al.*, 2014; *Shi et al.,* 2017].
Hiss intensity modulation is often driven by the variation of background plasma
density either through local amplification or wave propagation [*Chen et al.*, 2012c], and
the modulation of hiss by other factors may easily be suppressed by the effect of the plasma
density. Therefore, observations showing direct correlation between hiss emission and
electron flux are still very limited. In fact, electron fluxes of energetic electrons (tens to
hundreds of keV) can be modulated by Ultra Low Frequency (ULF) waves. A typical
modulation is caused by drift-resonance [*Southwood and Kivelson*, 1981]. *Zong et al.*
[2009] showed an interesting event of energetic electron modulation by shock induced ULF
waves. More recently, *Claudepierre et al.* [2013] presented observations of electron drift
resonance with the fundamental poloidal mode of ULF waves based on Van Allen Probes
measurements. The energy dependence of the amplitude and phase of the electron flux
modulations provided strong evidence for such an interaction. The peak electron flux
modulations occurred over 5-6 wave cycles at energies ~ 60 keV. The drift-resonance
between electrons and ULF waves has been extensively studied both theoretically and
observationally based on Van Allen Probes data [*Dai et al.*, 2013; *Hao et al.*, 2014; *Chen*
*et al.*, 2016; *Zhou et al.*, 2015, 2016; *Li et al.*, 2017]. Such modulation of energetic electrons
may modulate hiss emissions by varying the electron flux and pitch angle anisotropy, which
could potentially affect the local growth rates of hiss waves, but the observational evidence
has not been reported yet. In this study, we report on a modulation of hiss wave intensity
and injected electron flux due to ULF waves observed by Van Allen Probe B near the
dayside, providing clear evidence that the hiss emission was generated through local
amplification in the outer plasmasphere.

## 105 **2. Data and Methodology**

The Van Allen Probes comprise two identical spacecraft (Probes A and B) in near-
equatorial orbits with an altitude of ~600 km at perigee and geocentric distance of ~5.8 $R_E$
at apogee [*Mauk et al.*, 2012]. The Electric and Magnetic Field Instrument Suite and
Integrated Science (EMFISIS) suite on Van Allen Probes A and B includes a magnetometer
and a Waves instrument [*Kletzing et al.*, 2013]. The DC magnetic field is measured by the
magnetometer, and the survey mode of Waveform Receiver (WFR) provides the power
spectral density from 10 Hz to 12 kHz at 6 s time resolution. Plasma density can be either
calculated based on the upper hybrid resonance frequency extracted from the High
Frequency Receiver (HFR) data [*Kurth et al.,* 2015] or be inferred from the spacecraft
potential measured by the Electric Field and Waves (EFW) instrument [*Wygant et al.,*
2013]. We inferred plasma density profiles based on the measurements from both
instruments in the present study to obtain accurate plasma density values with high time
resolution. High resolution electron flux measurements over the energy range of ~30 keV
to 4 MeV are provided by the Magnetic Electron Ion Spectrometer (MagEIS) instrument
[*Blake et al.*, 2013; *Spence et al.*, 2013]. We used the level 3 MagEIS dataset which
includes particle pitch angle distribution in this study to evaluate the electron distribution
responsible for the hiss wave generation.

## 3. Observational Results

A hiss intensification event modulated by electron injection was observed by Van
Allen Probe B during ~20-22 UT on 12 January 2014, as shown in Figure 1. The satellite
was located on the dayside and remained inside the plasmasphere, indicated by the high
plasma density (Figure 1f). The main power of the hiss emission (Figures 1b and 1c)
resided below the lower hybrid resonance frequency (white dash-dotted line in Figure 1b)
and 100 Hz (white dashed line in Figure 1c) and intensified following the increase in the
AE index (Figure 1a). Figure 1e presents the magnitude of the background magnetic field.
The spin averaged electron flux (Figure 1g) exhibited modulations with a period of about
6 minutes. There is also a variation in the electron pitch angle anisotropy (Figure 1h)
although it is not as clear as the modulations of electron flux. The electron anisotropy is
calculated based on *Chen et al*. [1999]. The black lines in Figures 1g and 1h show the
calculated minimum electron resonant energy for the first-order cyclotron resonance with
parallel-propagating right-hand polarized waves at a frequency of 40 Hz (magenta line in
Figure 1b). As shown in Figure 1g, the minimum resonant energy captures the main energy
of injected electrons. Figure 1i shows the electron pitch angle distribution at 54 keV which
exhibits a pronounced modulation. The vertical dashed lines present the minima of the
electron fluxes at 54 keV. Figure 1d illustrates the convective linear growth rates for
parallel-propagating whistler mode waves that were calculated using the electron
distribution measured by MagEIS based on the equations of *Summers et al.* [2009]. The
modulation of linear growth rate appears to correlate well with the observed hiss wave
spectral intensity with a period of several minutes.

Changes in the background magnetic field, plasma density and the injected electron

distribution (flux and pitch angle anisotropy of resonant electrons) could potentially be
responsible for the hiss wave growth. Since the variation of the background magnetic field
is small ($\sim$ 4 nT) compared to the median value ($\sim$ 150 nT), the effect of background
magnetic field on the wave growth rate is likely to be insignificant compared to the effects
of plasma density and electron injection. To distinguish the roles of these two effects in the
local wave amplification, we compared the hiss wave amplitude with spin averaged
electron flux and plasma density. The hiss wave amplitude integrated from 20 Hz to 1000
Hz is shown in Figure 2a. Figure 2b presents the spin averaged electron flux integrated
over the energy range from 30 keV to 200 keV. The vertical dashed lines in Figure 2 depict
the same times as in Figure 1.

Figure 2c shows the comparison between the filtered electron flux (black) over 1.5

mHz - 4 mHz and the filtered hiss wave intensity (blue) over 1.5 mHz - 4 mHz. It suggests
that the hiss intensity is well correlated with the variation of the electron flux. The
correlation coefficient between the filtered electron flux and the filtered hiss wave intensity
in the time period from 20:00 UT to 22:00 UT is 0.841. The satellite was located at a
magnetic latitude of -1.3°~-2.0°, which was near the source region where local wave
amplification typically occurs, and this is probably why hiss intensity and electron flux
exhibit a remarkable correlation.

In the present hiss modulation event, the filtered background plasma density (green

line in Figure 2d) is not well correlated with the filtered wave intensity (with a correlation
coefficient of 0.105), especially during the period from 20:45 UT to 21:40 UT. This
suggests that the variation of plasma density plays an insignificant role in the modulation
of hiss wave intensity during this event. To investigate the sole effect of density on hiss
intensity, we also calculated the correlation coefficient between the non-filtered hiss wave
intensity and non-filtered the plasma density which even shows a slight anti-correlation
with a coefficient of ~ -0.483.

The comparison between the filtered electron pitch angle anisotropy at 54 keV and

filtered wave intensity is shown in Figure 2e. Although a correlation coefficient of 0.378
indicates a certain correlation between these two parameters, it is much lower than the
correlation between the hiss wave intensity and electron flux (0.841). Therefore, we
suggest that the variation of electron pitch angle anisotropy play a less important role in
hiss intensity modulation compared to the variation of electron flux.
The electron flux variation observed by Van Allen Probe B may be caused by ULF
wave modulation since they have similar time periods. Figure 3 shows the variation of
electron fluxes at different energy channels observed by both Van Allen Probe A (a) and
Van Allen Probe B (b). At ~19:30UT, both probes, especially Van Allen Probe B observed
intense electron injections. Between 20:00 and 22:00 UT, the energetic electron fluxes
observed by Probe B are modulated at most energy channels, with a time period of several
minutes in the same frequency range as typical ULF waves (Pc4-5).
Figure 4 is the summary of the Pc4-5 ULF waves from Van Allen Probe B during the
time interval of interest (20:00–22:00 UT). Dynamic spectrograms of the ULF wave
powers are shown for the three components of the magnetic field (in the mean field-aligned,
geocentric solar magnetospheric (GSM) coordinates) along with the y component of the
electric field in modified geocentric solar elliptic (MGSE) coordinate. Band-pass filtered
time series (1.5-4 mHz) are shown below for each dynamic spectrogram. The parallel
magnetic field ($B_{para}$) and y component electric field in MGSE coordinate ($E_y$) have a
similar frequency peak at ~2.6 mHz. The wave spectra of the $E_y$ and $B_{para}$ components
suggest that the compressional mode and shear mode are likely coupled.
The correlation of the ULF waves and the energetic electron fluxes at different energy
channels is shown in Figure 5. Figure 5a illustrates the filtered $E_y$ component of the electric
field between 1.5 and 4 mHz. Since Van Allen Probe B is near noon, the $E_y$ component
approximately represents the electric field in the azimuthal direction. Band-pass filtered
electron fluxes normalized by unperturbed levels at different energy channels are shown in
Figure 5b. The vertical black lines indicate the minima of the $E_y$ component. The electron
fluxes at various energies show a modulation period which is very similar to that of $E_y$.
Besides, these fluxes exhibit an energy-dependent phase shift with respect to $E_y$. The phase
of the electron flux oscillations with respect to $E_y$ is closest to 180° out-of-phase at ∼ 466
keV. At lower energies, the phase of peak electron fluxes relative to the $E_y$ minimum varies
but is not 180° out-of-phase. For the observed modulating hiss, the minimum resonant
energy is tens of keV (Figure 1), and thus the electron flux at energy below 100 keV plays
a dominant role in hiss amplification. Although these low energy electrons (30–100 keV)
are not exactly in drift resonance with the observed ULF waves, their modulation is highly
relevant to the presence of ULF waves. These low energy electrons may be accelerated by
the ULF waves during the first half cycle and then decelerated so that there is no total
energy gain. This mechanism was also demonstrated in the drift-resonance theory in which
the peak electron fluxes should have a 180° energy shift [*Southwood and Kivelson*, 1981].

Meanwhile, Van Allen Probe A detected hiss emissions in a similar frequency range

as shown in Figure 6. During this time period, Van Allen Probe A was located at lower $L$
shells (2.6 < L < 5.3) and later MLTs (14.9 < MLT < 18.0). The hiss intensity also exhibited
modulation in electric and magnetic field, as shown in Figures 6b and 6c, respectively.
However, different from the observation by Probe B, the hiss intensity is dominantly
modulated by the variation of the plasma density. Figure 6d shows the density profile
obtained from EMFISIS (black) and EFW (red). Examples of evident modulations by
variation of plasma density are highlighted with grey blocks. According to ray tracing
simulation [*Chen et al.*, 2012c], the hiss waves tend to propagate to the region with higher
density resulting in higher wave intensity. Figures 6e and 6f show the spin averaged
electron flux and pitch angle anisotropy based on MagEIS data and the white lines are the
minimum resonant energy corresponding to a frequency of 40 Hz (Figure 6b). There is no
clear correlation between the hiss intensity and electron flux, suggesting that the
modulations are mainly caused by the plasma density variation. We also calculated the
convective linear growth rates for parallel-propagating whistler mode waves as shown in
Figure 6g. The growth rate profile shows little correlation with that of the observed hiss
intensity, indicating that these waves are not locally excited.
Figure 7 illustrates the comparison of hiss wave frequency spectra observed by Van
Allen Probes A (Figures 7a-7b) and B (Figures 7c-7d). At the beginning of the emission
around 20:20 UT, the hiss wave intensity as a function of frequency observed by Van Allen
Probe A presents a minimum at ~200 Hz (indicated by the white arrows in Figures 7a and
7b). This feature is similar to the observation by Van Allen Probe B (Figures 7c and 7d),
where the modulation of hiss wave power below 100 Hz is correlated with the calculated
wave growth rate (Figure 1d) based on the observed electron distribution. The hiss wave
frequency spectra and structures observed by Probe A are similar to those observed by
Probe B, but the energy spectra of energetic electrons are significantly different. Therefore,
the hiss emission observed by Probe A may be the result of wave propagation from the
source region in the outer plasmasphere and further modulated by the local plasma density
variation.

## 4. Summary and Discussion
We report clear evidence of local amplification of plasmaspheric hiss observed by Van
Allen Probe B in the postnoon sector of the outer plasmapshere. The minimum resonance
energy calculated for the observed hiss wave frequency is consistent with the energy of
injected electrons. The hiss wave intensity was modulated by the injected energetic
electrons, which were modulated by ULF waves. In the meantime, Van Allen Probe A also
observed similar hiss emissions at lower $L$ shells, which is probably due to the propagation
from the source region in the outer plasmasphere. Different from the observation by Probe
B, the hiss wave intensity observed by Probe A is predominantly affected by the
background plasma density. The modulation of hiss intensity by plasma density could be
due to the effect of ray focusing at high-density region during propagation [*Chen et al.*,
2012c].
Figure 8 summarizes the processes discussed in this study. The injected energetic
electrons with energies of tens to hundreds of keV drift from the nightside to the dayside
in the outer plasmasphere. Simultaneously, the ULF waves modulate the energetic electron
fluxes. The modulated energetic electrons then lead to the modulation of the hiss intensity
via local amplification. These features were all well captured by Van Allen Probe B. During
the same time period, Probe A at a later MLT and lower $L$ shell observed hiss emissions
which may originate from the source region in the outer plasmasphere.
Chorus waves which are intense coherent electromagnetic emissions exhibiting discrete
rising or falling tones are believed to be generated through cyclotron resonance with
anisotropic electrons [*Kennel and Petschek*, 1966; *Anderson and Maeda*, 1977; *Meredith*
*et al.*, 2001; *Li et al.,* 2009]. It has been shown that ULF waves can modulate chorus
intensity by modulating the background magnetic field and/or plasma density which affect
the number of energetic electrons resonant with chorus waves [*Li et al.,* 2011]. Besides, the
ULF wave-induced modulation of chorus could have an impact on electron precipitation
leading to pulsating aurora [*Jaynes et al.*, 2015]. Similar modulations may also be captured
in hiss wave intensity if hiss is locally amplified. However, different from chorus,
plasmaspheric hiss waves are commonly known to be structureless [*Thorne et al.*, 1973]
and wave propagation is believed to be important for the measured hiss wave
intensification [*Bortnik et al.*, 2008, 2009; *Chen et al.*, 2014]. The hiss wave intensity is
typically modulated by the variation of the background plasma density [*Chen et al.,* 2012c].
Nonetheless, our study showed the first evidence of the hiss wave modulation caused by
modulated injected electrons due to ULF waves, clearly indicating that the hiss is locally
amplified in the outer plasmasphere. It also provides an interesting link between the ULF
waves and hiss waves which are in two distinct frequency ranges but both play important
roles in radiation belt electron dynamics.

## Acknowledgments

**Acknowledgments**
The work at Boston University is supported by the NASA grants NNX15AI96G,
NNX17AG07G, and NNX17AD15G and the NSF grant AGS-1723342. The research at
the University of Minnesota was supported by JHU/APL contract UMN 922613 under
NASA contract JHU/APL NAS5-01072. We acknowledge the RBSP-ECT and EMFISIS
funding provided by JHU/APL contract No. 967399 and 921647 under NASA's prime
contract No. NAS5-01072. We would like to thank Dr. Lei Dai and Dr. Xu-Zhi Zhou for
very helpful discussions in this study. We would like to acknowledge the EMFISIS data
obtained from http://emfisis.physics.uiowa.edu, the MagEIS data obtained from
http://www.rbsp-ect.lanl.gov/science/DataDirectories.php, and the EFW data obtained
from http://rbsp.space.umn.edu/data/rbsp/. We also thank the World Data Center for
Geomagnetism, Kyoto for providing AE index used in this study.

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

## Figure Captions

**Figure 1.** Plasmaspheric hiss modulation caused by injected electrons observed by Van Allen Probe B from 20:00 UT to 22:00 UT on January 12, 2014. (a) *AE* index; frequency-time spectrogram of (b) wave electric field and (c) wave magnetic field spectral density in the WFR channel; (d) frequency spectrum of convective linear wave growth rates; (e) background magnetic field intensity; (f) calibrated plasma density based on EFW and EMFISIS; (g) spin-averaged electron flux measured by MagEIS; (h) electron pitch angle anisotropy; (i) pitch angle distribution of electrons at 54 keV. The white dash-dotted line in Figure 1b represents the lower hybrid resonance frequency ($f_{LHR}$). The magenta line in Figure 1b indicates 40 Hz. The white dashed line in Figure 1c indicates 100 Hz. The black lines in Figures 1g and 1h represent the minimum resonant energy of electrons interacting with the waves at 40 Hz. The dashed vertical lines mark the modulation of the electron flux at 54 keV (Figure 1i).

**Figure 2.** (a) Integrated hiss intensity from 20 Hz to 1000 Hz; (b) integrated spin-averaged electron flux from 30 keV to 200 keV; (c) filtered integrated electron number flux (black) and filtered magnetic wave intensity of hiss (blue); (d) filtered plasma density (green) and filtered magnetic wave intensity of hiss (blue); (e) filtered pitch angle anisotropy (red) and filtered magnetic wave intensity of hiss (blue). The vertical dashed lines depict the same times as those in Figure 1.

**Figure 3.** Variation of electron fluxes at different energies observed by Van Allen Probe A (a) and Van Allen Probe B (b). In Figure 3b, the modulation of electron fluxes was

observed by Van Allen Probe B between 20:00:00 and 22:00:00 UT in association with
ULF waves, and the dispersed electron injection was observed at ~19:30:00 UT.
**Figure 4.** Summary of the Pc4-5 ULF wave frequency spectra from Van Allen Probe B
during the time interval of interest (20:00-22:00 UT). Dynamic spectrograms are shown
for the three components of the magnetic field (in the mean field-aligned, GSM coordinates)
along with the y component of the electric field in MGSE coordinate. Band-pass filtered
time series (1.5 - 4 mHz) are shown below for each dynamic spectrogram. The black dashed
lines indicate the frequency at ~2.6 mHz.
**Figure 5.** The correlation of the filtered (1.5 - 4 mHz) $E_y$ component of ULF waves (a) and
the energetic electron fluxes at different energy channels (b). The electron fluxes show the
modulation in the similar period to that of $E_y$, but exhibit an energy-dependent phase shift
with respect to $E_y$.
**Figure 6.** The observation of waves and electron fluxes by Van Allen Probe A during the
same period as that in Figure 1. a) *AE* index; (b) frequency-time spectrogram of wave
electric field and (c) wave magnetic spectral density in the WFR channel; (d) plasma
density obtained by EFW (red) and EMFISIS (black); (e) spin-averaged electron flux
measured by MagEIS; (f) electron pitch angle anisotropy; (g) convective wave growth rates.
Grey block areas indicate the intervals of hiss modulation by variation of plasma density.
The magenta line in Figure 6b indicates 40 Hz. The black dashed line in Figure 6c indicates
100 Hz. The white lines in Figures 6e and 6f represent the minimum resonant energy of
electrons for the waves at 40 Hz.
**Figure 7.** The wave electric (a) and magnetic (b) spectral density observed by Van Allen
Probe A and the wave electric (c) and magnetic (d) spectral density from Van Allen Probe
B. Note that at the beginning of the emissions around 20:20 UT, the hiss wave intensity as
a function of frequency presents a minimum at ~200 Hz (white arrows) for the observations
from both Van Allen Probes A and B.
**Figure 8.** A cartoon showing energetic electron trajectory (green), ULF waves (pink) and
hiss intensity modulation (blue). Injected electrons from the nightside drift to the postnoon
sector (green arrow) in the outer plasmasphere where they provide a source of free energy
for hiss wave generation in the outer plasmasphere. During the period of electron injection,
electrons are modulated by ULF waves (magenta), which lead to the modulation of hiss
wave amplification (blue), as observed by Van Allen Probe B. The hiss waves are probably
generated in the outer plasmasphere, and then propagate into lower $L$ shells, as observed
by Van Allen Probe A.

Figure 1

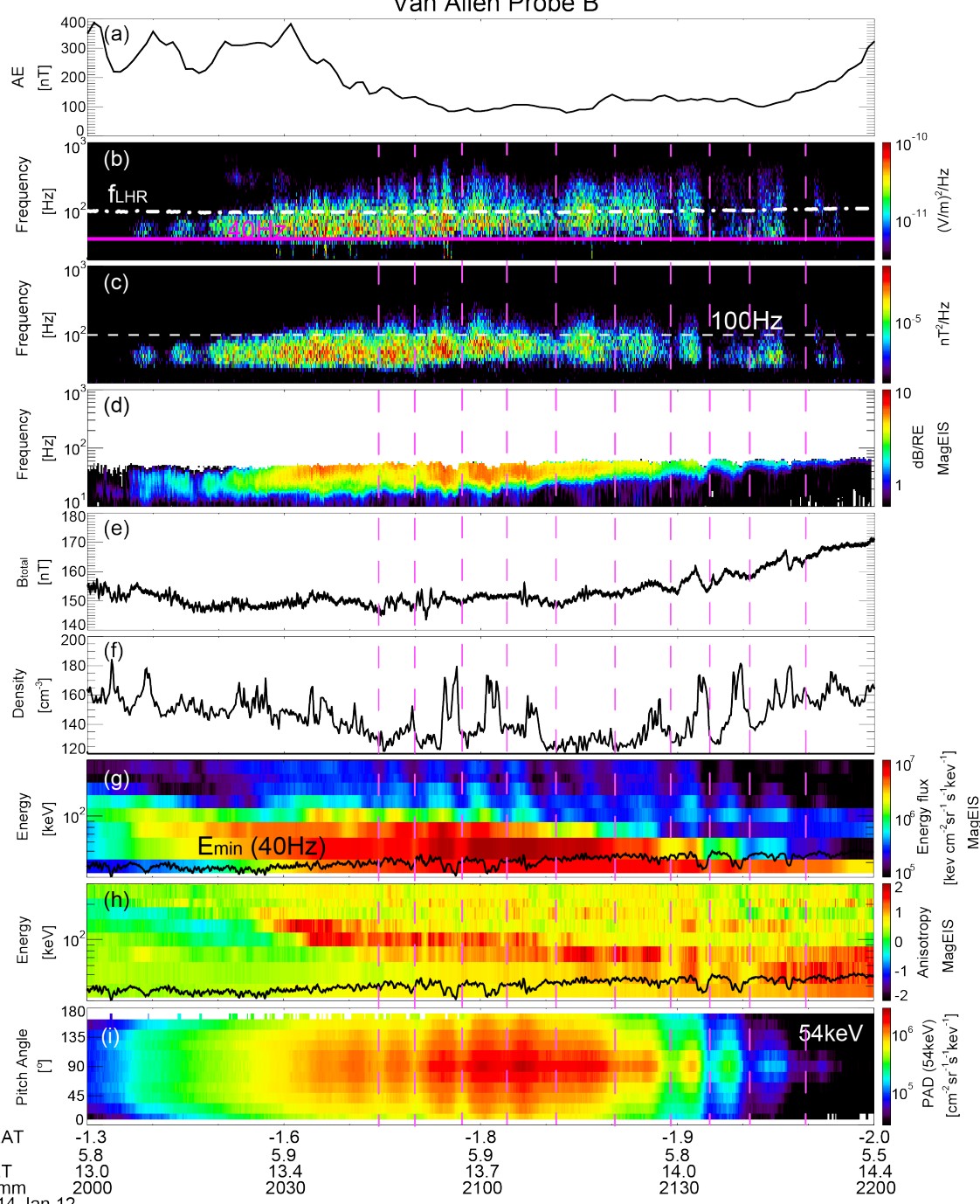



Figure 2

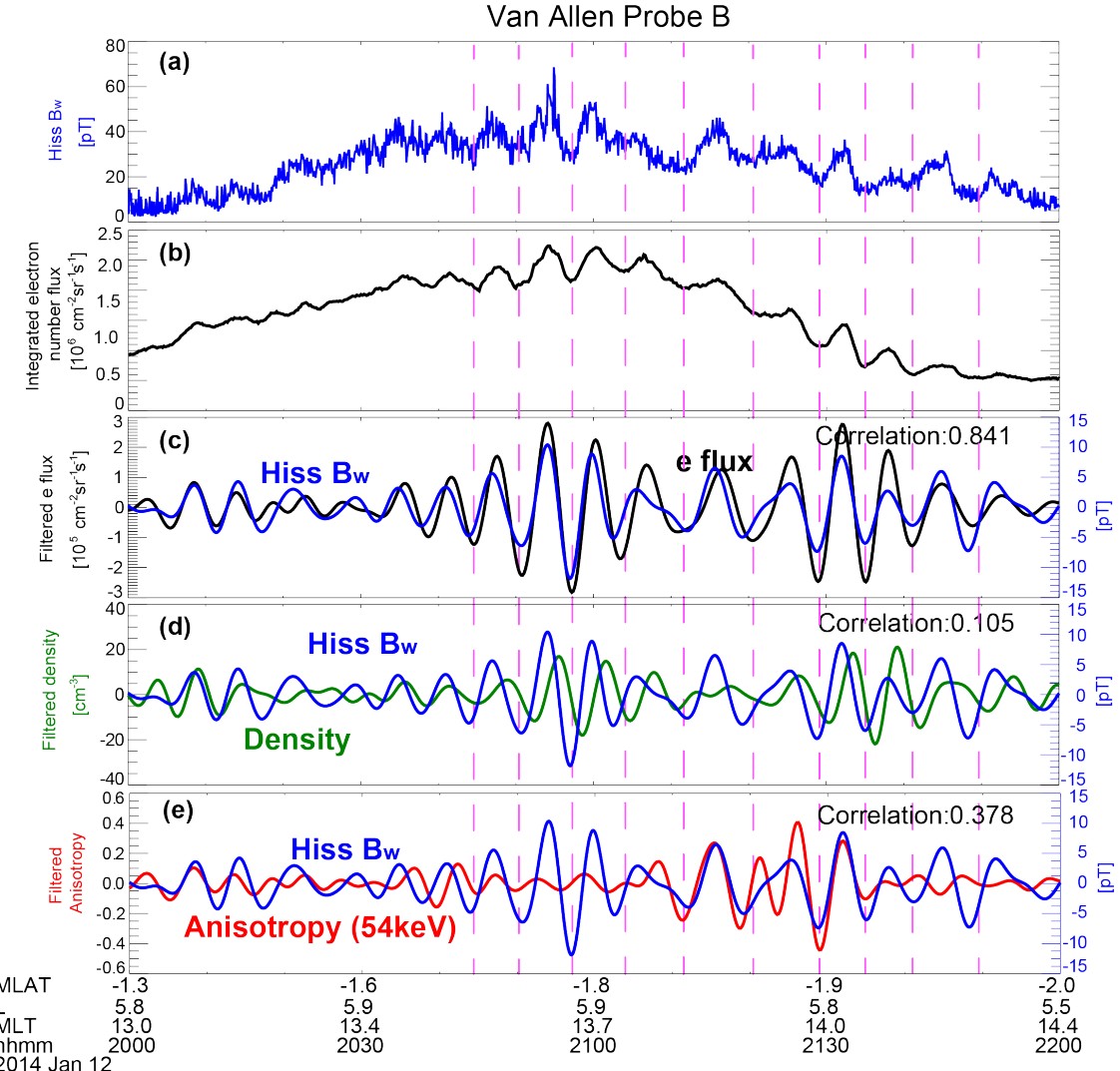



Figure 3.

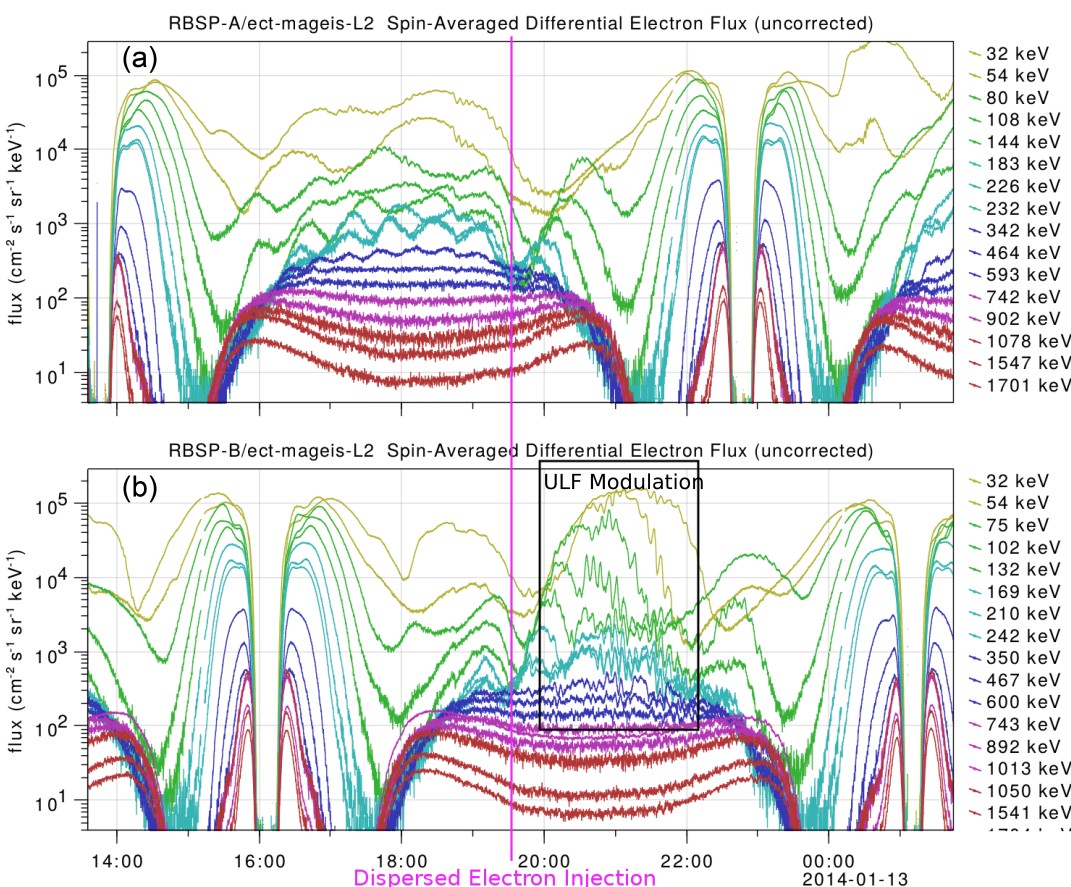




Figure 4

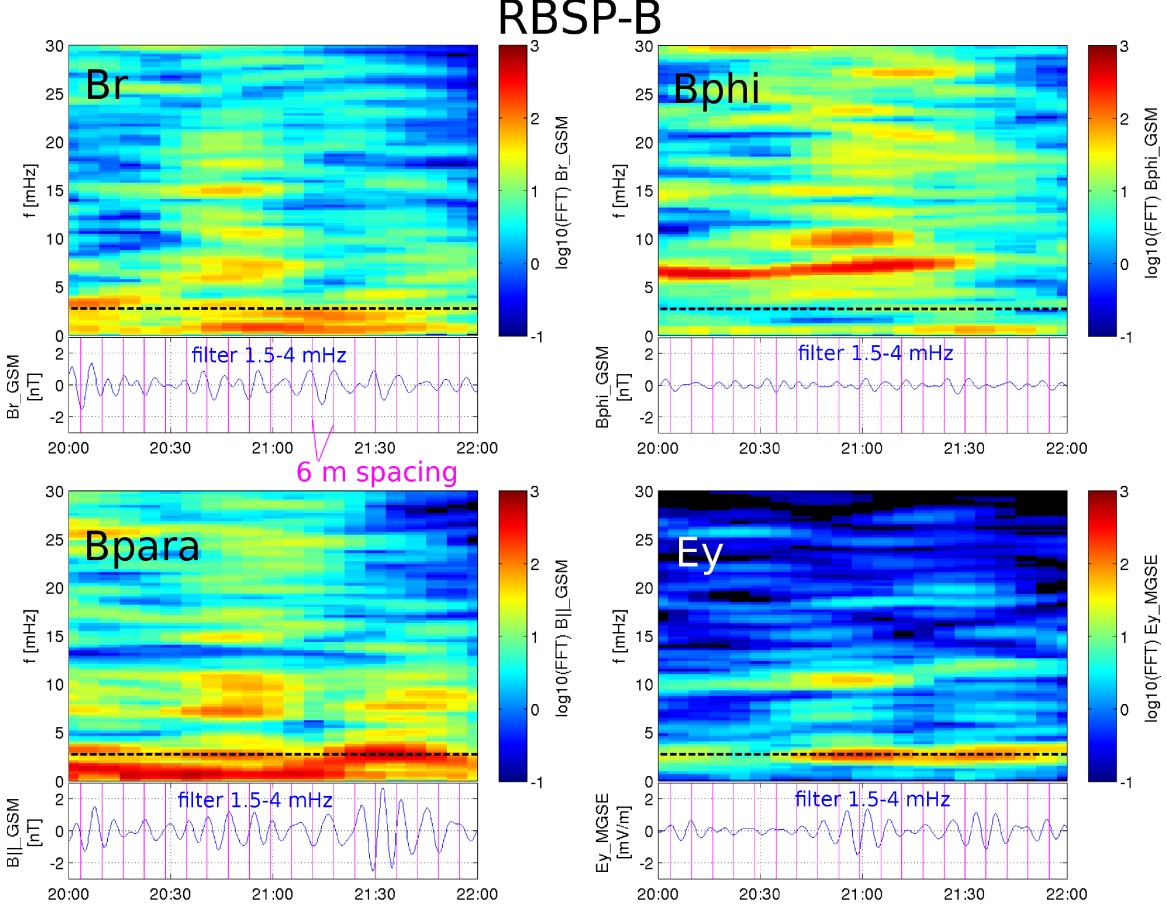



Figure 5

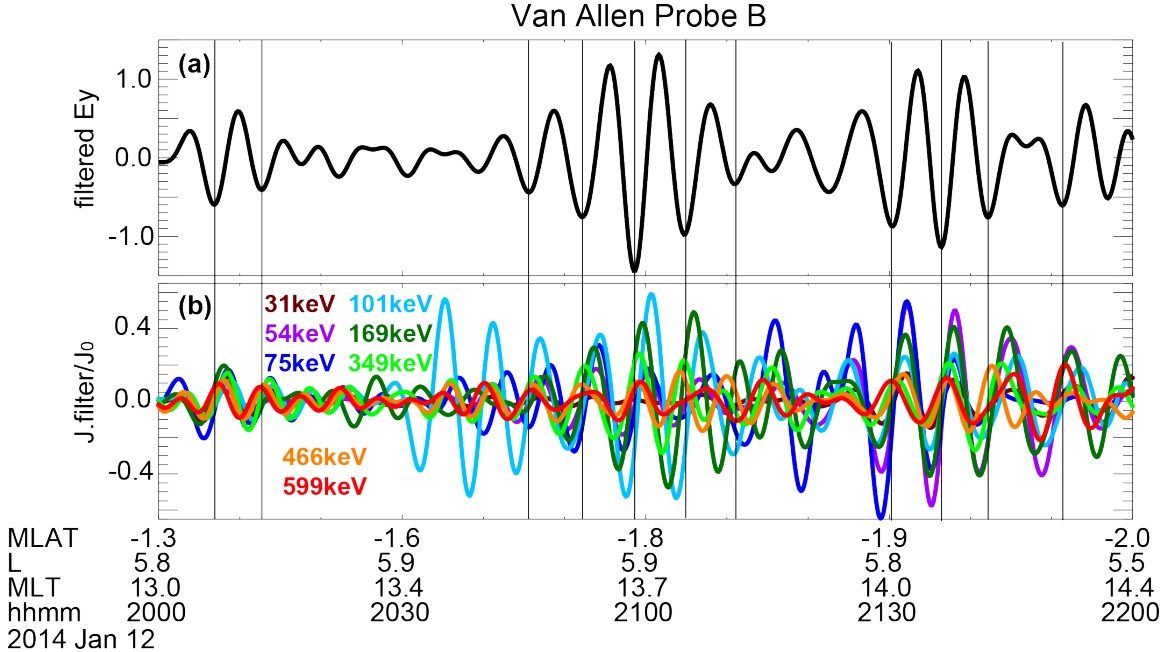




Figure 6

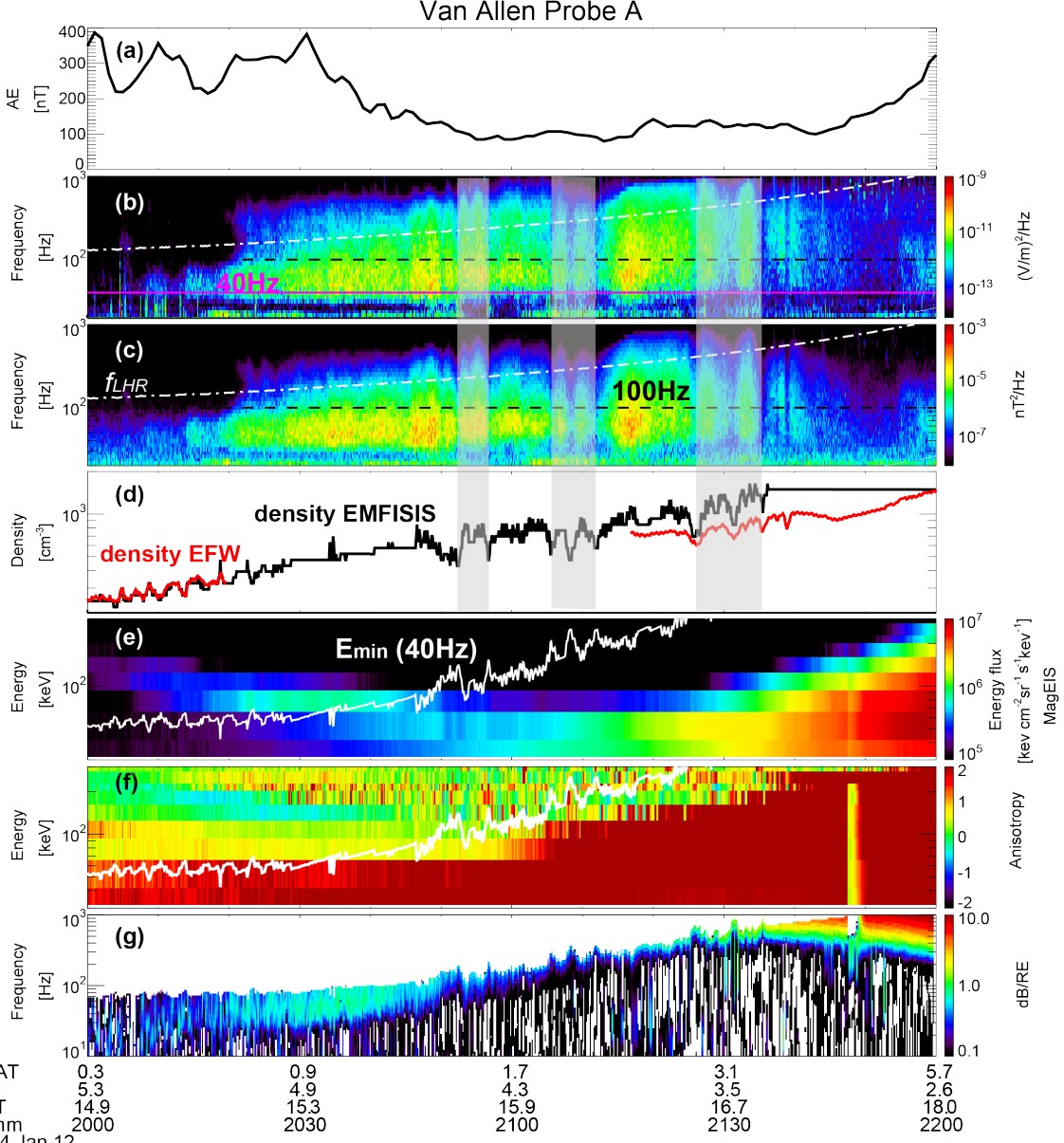



Figure 7

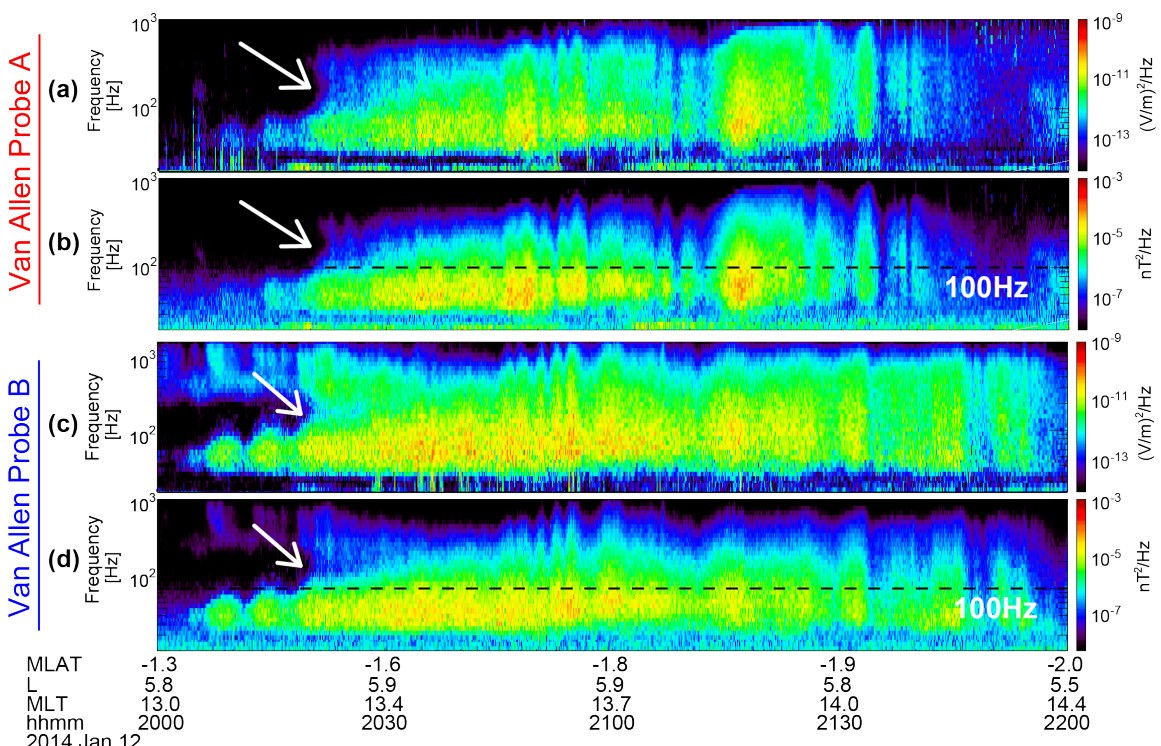


Figure 8

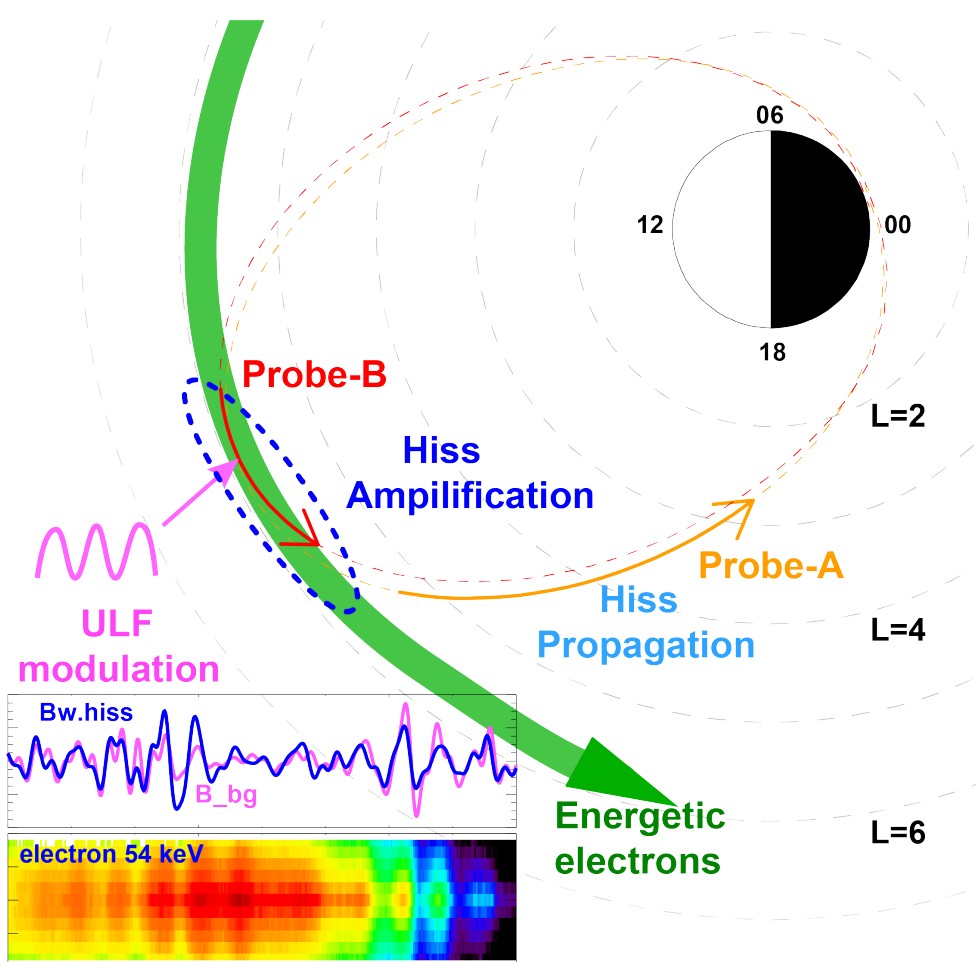
