# Peer review of "Van Allen Probes observation of plasmaspheric"

_Annales Geophysicae, 2018_

## Referee Comment (RC1) · Anonymous Referee #1 · 7 Feb 2018

Evaluation:

The authors present and interesting and compelling study showing the modulation of hiss wave intensity and injected electron flux by ULF waves observed by Van Allen probe B near the dayside. The results provide important evidence that the hiss intensities observed were likely generated by local amplification in the outer plasmasphere. In contrast, the hiss emissions observed by Van Allen probe A, at lower L shells, were not associated with electron injections and were primarily modulated by plasma density. The Van Allen probe A measurements suggest that the hiss observed deep inside the plasmasphere may have propagated from higher L shells. This is a very good and comprehensive study and I have no hesitation in recommending it for publication in Annales Geophysicae. I have a small number of minor comments that the authors may

wish to take into account when preparing the next draft of their paper.

Minor Comments:

Line 49. The authors should consider including references to Meredith et al. [2007, 2009] who also demonstrated the importance of plasmaspheric hiss in the loss of energetic electrons in the plasmasphere

References:

Meredith, N.P., Horne, R. B., Glauert, S. A., & Anderson, R. R. (2007), Slot region electron loss timescales due to plasmaspheric hiss and lightning generated whistlers, Journal of Geophysical Research, 112, A08214, doi:10.1029/2006JA012413

Meredith, N. P., Horne, R. B., Glauert, S. A., Baker, D. N., Kanekal, S. G., & Albert, J.M. (2009), Relativistic electron loss timescales in the slot region, Journal of Geophysical Research, 114, A03222, doi:10.1029/2008JA013889

Line 200. Please explain in a little more detail why the modulation of the low energy electrons is highly relevant to the presence of ULF waves.

---

## Referee Comment (RC2) · Anonymous Referee #2 · 21 Feb 2018

Review of "Van Allen Probes observation of plasmaspheric hiss modulated by injected energetic electrons" by Shi et al. for Annales Geophysicae

The results described on this manuscript are of relevance to the community and interesting. It provides evidence, coming from one event, where local amplification of hiss waves is likely as opposed to other sources such as chorus waves. The reasoning and the analysis done are sound and complete for the most part. The manuscript is also mostly well-written and clear. I have a few suggestions to help make the paper more clear and a few minor points that should be addressed.

Minor points:

l. 149 – In the beginning of the paragraph, four parameters are mentioned (background,

magnetic field, plasma density, electron flux and electron pitch-angle anisotropy) and after discarding one of them, you continue to talk about "these two effects" meaning the plasma density and the electron flux in figure 2, but not the anisotropy. Why is the anisotropy being dropped from the subsequent analysis? Maybe the analysis should be done and briefly described, even if the results are not interesting or a justification for not doing the analysis should be given.

l. 195 – It is possible to estimate (using a dipole) the drift frequency of the electrons at this energy of 466keV, does it match with the ULF frequency measured by the azimuthal component of the electric field? In other words, are the electrons at these energies in drift resonance with these waves? Doing a calculation (which should be double-checked) of the drift period found on Roederer's book, I found the period for this energy to be ∼21 minutes. If this is not a drift resonance effect, why is the flux being modulated? Perhaps, it is more related to the plasmasheet population that was freshly injected, but some sort of reasoning should be given for the correlation between ULF waves and oscillations of low-energy electron fluxes.

Clarifications:

In the sentence starting in line 73, it is not clear if you are saying that low frequency hiss causes more changes to the electron pitch angle distribution than normal hiss or if it just causes some changes in the electron pitch angle distribution.

l.87 – "...energetic electrons can be modulated by ULF wave". Here and in other parts of the introduction, I assume you mean flux modulation, but it should be clarified.

l. 98 – Here and throughout the text you mention electron anisotropy. Again, I assume you mean the pitch angle anisotropy, but it should be clarified, because others unfamiliar with this specific topic may think you are referring to temperature anisotropy or pressure anisotropy.

l.131 – It should be noted that the modulation is not very clear, specially without the

guiding vertical lines, in the electron pitch angle anisotropy (Figure 1h). I can see some oscillations, but not with the period of 6 minutes as claimed here. I think the panel should be kept for reference but the description of the modulation should be different from the description of the electron flux (Figure 1g).

---

## Author Comment (AC1) · 22 Mar 2018

We thank both reviewers for their careful reading of the manuscript and valuable comments for improvements. We have made point-by-point responses to the detailed comments by both reviewers indicated in the blue color, and included a new version of the manuscript with highlighted changes.

Responses to Reviewer #1:

Evaluation: The authors present and interesting and compelling study showing the modulation of hiss wave intensity and injected electron flux by ULF waves observed by Van Allen probe B near the dayside. The results provide important evidence that the hiss intensities observed were likely generated by local amplification in the outer
plasmasphere. In contrast, the hiss emissions observed by Van Allen probe A, at lower L shells, were not associated with electron injections and were primarily modulated by plasma density. The Van Allen probe A measurements suggest that the hiss observed deep inside the plasmasphere may have propagated from higher L shells. This is a very good and comprehensive study and I have no hesitation in recommending it for publication in Annales Geophysicae. I have a small number of minor comments that the authors may wish to take into account when preparing the next draft of their paper.

Reply: The authors would like to thank Reviewer #1 for the positive evaluation and helpful comments to improve the paper quality. We have made the following responses to the reviewer's specific comments.

Minor Comments: Line 49. The authors should consider including references to Meredith et al. [2007, 2009] who also demonstrated the importance of plasmaspheric hiss in the loss of energetic electrons in the plasmasphere References: Meredith, N.P., Horne, R. B., Glauert, S. A., & Anderson, R. R. (2007), Slot region electron loss timescales due to plasmaspheric hiss and lightning generated whistlers, Journal of Geophysical Research, 112, A08214, doi:10.1029/2006JA012413 Meredith, N. P., Horne, R. B., Glauert, S. A., Baker, D. N., Kanekal, S. G., & Albert, J.M. (2009), Relativistic electron loss timescales in the slot region, Journal of Geophysical Research, 114, A03222, doi:10.1029/2008JA013889

We thank the reviewer for the suggestion. We have added these two references in the manuscript (see Line 49 of the revised manuscript with highlighted changes).

Line 200. Please explain in a little more detail why the modulation of the low energy electrons is highly relevant to the presence of ULF waves.

We thank the reviewer for the suggestion. We have added the discussion in the manuscript (see Lines 209-212):

"These low energy electrons may be accelerated by the ULF waves during the first

half cycle and then decelerated so that there is no total energy gain. This mechanism was also demonstrated in the drift-resonance theory in which the peak electron fluxes should have a 180° energy shift [Southwood and Kivelson, 1981]."

Please also note the supplement to this comment:
https://www.ann-geophys-discuss.net/angeo-2018-2/angeo-2018-2-AC1-supplement.pdf

**Supplement:**

[revised manuscript text omitted]

Figure 2

[Figure]

Figure 3.

[Figure]

Figure 4

[Figure]

Figure 5

[Figure]

Figure 6

[Figure]

Figure 7

[Figure]

Figure 8

[Figure]

---

## Author Comment (AC2) · 22 Mar 2018

We thank both reviewers for their careful reading of the manuscript and valuable comments for improvements. We have made point-by-point responses to the detailed comments by both reviewers indicated in the blue color, and included a new version of the manuscript with highlighted changes.

Responses to Reviewer #2:

The results described on this manuscript are of relevance to the community and interesting. It provides evidence, coming from one event, where local amplification of hiss waves is likely as opposed to other sources such as chorus waves. The reasoning and the analysis done are sound and complete for the most part. The manuscript is also Printer-friendly version

mostly well-written and clear. I have a few suggestions to help make the paper more clear and a few minor points that should be addressed.

Reply: The authors would like to thank Reviewer #2 for the positive evaluation and helpful comments to improve the paper quality. We have made the following responses to the reviewer's specific comments.

Minor points: I. 149 – In the beginning of the paragraph, four parameters are mentioned (background, magnetic field, plasma density, electron flux and electron pitchangle anisotropy) and after discarding one of them, you continue to talk about "these two effects" meaning the plasma density and the electron flux in figure 2, but not the anisotropy. Why is the anisotropy being dropped from the subsequent analysis? Maybe the analysis should be done and briefly described, even if the results are not interesting or a justification for not doing the analysis should be given.

We thank the reviewer for this comment. We have added a panel (e) to Figure 2 which shows the comparison between the filtered hiss wave intensity and the filtered electron pitch angle anisotropy. It demonstrates some correlations but not as high as that between electron flux and wave intensity. Thus, we suggest that the variation of pitch angle anisotropy plays a minor role. We have added this point in the manuscript (see Lines 173-178 of the revised manuscript with highlighted changes).

I. 195 – It is possible to estimate (using a dipole) the drift frequency of the electrons at this energy of 466keV, does it match with the ULF frequency measured by the azimuthal component of the electric field? In other words, are the electrons at these energies in drift resonance with these waves? Doing a calculation (which should be double-checked) of the drift period found on Roederer's book, I found the period for this energy to be 21 minutes. If this is not a drift resonance effect, why is the flux being modulated? Perhaps, it is more related to the plasmasheet population that was freshly injected, but some sort of reasoning should be given for the correlation between ULF waves and oscillations of low-energy electron fluxes.
We thank the reviewer for this comment. We suggest that the electrons at energy of 690keV could be in drift resonance with the ULF waves, since they have a similar frequency and the flux at 690keV is 180° out-of-phase with the Ey component of the ULF waves. But at lower energies, which is the energy range of electrons that can generate the observed hiss intensity, the electrons are not in resonance with ULF waves. These electrons may be accelerated through non-resonant acceleration. For example, these electrons may be accelerated by the ULF waves during the first half cycle and then decelerated so that there is no total energy gain. This mechanism was also illustrated in the drift-resonance theory in which the peak electron fluxes should have a 180° energy shift. Following the reviewer's comment, we made the corresponding changes in the manuscript (see Lines 209-212).

Clarifications: In the sentence starting in line 73, it is not clear if you are saying that low frequency hiss causes more changes to the electron pitch angle distribution than normal hiss or if it just causes some changes in the electron pitch angle distribution.

We thank the reviewer for this comment. We meant to state that the low frequency hiss tends to increase the loss rate of energetic electrons (from  $\sim$ 50 keV to a few MeV) due to its stronger pitch angle scattering rates compared to the normal hiss. We have changed this sentence (see Lines 73-76):

"The low frequency hiss was demonstrated to cause more efficient loss of high energy electrons (from  $\sim$ 50 keV to a few MeV) due to its stronger pitch angle scattering rates compared to normal hiss [Ni et al., 2014; Li et al., 2015a]."

I.87 – "...energetic electrons can be modulated by ULF wave". Here and in other parts of the introduction, I assume you mean flux modulation, but it should be clarified.

We thank the reviewer for this suggestion. The reviewer is correct that we mean "flux modulation". We have made the corresponding changes throughout the manuscript.

I. 98 – Here and throughout the text you mention electron anisotropy. Again, I assume
you mean the pitch angle anisotropy, but it should be clarified, because others unfamiliar with this specific topic may think you are referring to temperature anisotropy or pressure anisotropy.

We thank the reviewer for this suggestion. The reviewer is correct that we mean "pitch angle anisotropy". We have revised the manuscript accordingly.

I.131 – It should be noted that the modulation is not very clear, specially without the guiding vertical lines, in the electron pitch angle anisotropy (Figure 1h). I can see some oscillations, but not with the period of 6 minutes as claimed here. I think the panel should be kept for reference but the description of the modulation should be different from the description of the electron flux (Figure 1g).

We thank the reviewer for this suggestion. We have modified the sentence to clarify this point (see Lines 132-134):

Please also note the supplement to this comment: https://www.ann-geophys-discuss.net/angeo-2018-2/angeo-2018-2-AC2supplement.pdf
**Discussion** paper

Fig. 1.